# "Education Is a Cultural Weapon": The Inner London Education Authority and the Politics of Literature for Young People

**Karen Sands-O'Connor**

School of English Literature, Language and Linguistics, Newcastle University, Newcastle-upon-Tyne NE1 7RU, UK; karen.sands-o'connor@ncl.ac.uk

**Abstract:** The Inner London Education Authority (ILEA) was founded in 1965 to manage education in London's inner boroughs; by the early 1970s, it was held up as one of the most progressive education experiments in British history. One of the marks of this progressiveness was its attention to London's Black child population and its attempts to connect with Black culture through multiculturalism. However, while the ILEA prided itself on its anti-racist, multicultural education methods, its publication arm often provided mixed messages about the value and place of Black students in the education system and society. Multiculturalism, which the ILEA used to guide the production of reading materials, often resulted in a lack of cultural specificity and an avoidance of issues facing Black students, such as racism. Partnering with Black educators allowed the ILEA to offer more culturally specific and anti-racist material, but doing so also brought the ILEA to the attention of critical governmental authorities, who would eventually disband the ILEA out of fear of Black radicalism.

**Keywords:** Inner London Education Authority; children's literature; anti-racism; Black British; Len Garrison; Ansel Wong; ACER

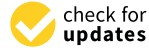



## 1. Introduction

> "Education is a cultural weapon and to black people, culture means political freedom."
>
> (Dhondy [1974] 2019, p. 42)

In 1965, the Greater London Council (GLC) took over from the London County Council as the body responsible for education in the British capital. In order to better manage the vast differences between central London's schools and the outer suburbs, the GLC created the Inner London Education Authority (ILEA) to manage the twelve inner London boroughs and the City of London. The ILEA was, as an education authority, one of the greatest educational experiments ever conducted in Britain, attempting to cover early childhood to adult education in a wide array of subjects through a variety of methods and media. The authority ran its own television centre from 1969 to 1977, funded two museums (the Victorian-era anthropological and zoological collections of the Horniman Museum and the Geffrye Museum of the Home), managed the Cockpit Theatre in Marylebone, and had a training facility for secondary students in biology at the London Zoo. William Stewart quotes John Bangs, a former head of the National Union of Teachers, who argued that the ILEA was unique because "There was a teacher centre for every subject, centres of expertise where teachers could go to share best practice. Nothing like that exists now. There was also groundbreaking work on class, sex and race; ILEA research centre was extraordinary" (Stewart 2015). The ILEA's progressive attitudes led to political activism; Gary McCulloch comments that the ILEA enabled a wide range of local activists from different backgrounds to take part in urban educational reform, both in setting the agenda for social and political change and in putting it into practise at a grass-roots level (McCulloch 2017, p. 1013).

Many of these activists were interested in the education of the Windrush generation, Afro-Caribbean migrants who came to Britain in the years following the Second World

War, and their British-born children. However, the policies pursued by the ILEA did not take a single political line. Concerns about "immigrant" children began to appear in the late 1960s, regarding their potential lack of success in school. This was borne out by the number of these children being placed in Educationally Subnormal, or ESN, classrooms, which, according to the ILEA's own report, were made up of nearly 30% immigrant children, despite them being a much smaller percentage of the school population (*The Education of Immigrant Pupils in Special Schools for Educationally Subnormal Children* 1968). Countless publications were devoted to the "West Indian problem" with titles such as "Commonwealth Children in Britain" (London Council of Social Service 1967), "The Education of West Indian Immigrant Children" (ILEA 1968), and "Education and the Immigrants" (National Association of Schoolmasters 1969); however, perhaps the most influential were the government reports, particularly the Plowden Report (Plowden 1967), which argued that "books used in schools should be re-examined" (Plowden 1967, p. 71) for "out-of-date attitude"' (Plowden 1967, p. 71). How the Plowden Reports were interpreted varied widely, not only across the country but within the ILEA itself. As Black educationalist Gus John commented, "The ILEA was supposed to be a unitary authority providing education in some kind of coherent manner; however, the lack of uniformity of approach between the various phases of the education system was quite incredible" (John 1990, pp. 61–62). Educationalists and Black community activists both attempted to address the needs of Black children via the English curriculum. Through a consideration of the critical writings of those involved in teaching Black children in the ILEA and the reading materials produced by the ILEA and its partner organizations, this essay will examine the sometimes-conflicting politics of activists within and outside of the ILEA and how their efforts affected Black children in London's schools between the late 1960s and the abolition of the ILEA in 1988.

**2. As Seen on TV: Multiculturalism and the ILEA's Educational Television Service**

One of the earliest initiatives of the ILEA was the development of the Educational Television Service (ETS), which broadcast programmes throughout London on a daily basis in a variety of subjects. By 1970, the ETS broadcast on a closed circuit to all 1400 ILEA schools and to some London universities as well (Grimshaw 2015). The ETS attempted to teach curricular material while also transmitting social values. Programmes produced included "Equal Rights" (1977), "Race for Survival" (1979; about unemployment among West Indians), and a series on "Teaching the Slow Learner" (1978), which offered guidance to students with special educational needs. The cultures and heritage of children in British schools were represented by offering programmes on India and the Caribbean as well as on "Reggae and Rasta" (1977). The ILEA had long advocated an embrace of multiculturalism, meant to celebrate the variety of ethnic backgrounds of students rather than colourblindness in schools. Peter Newsam, who in 1976 as Deputy Education Officer for the ILEA travelled to New York City, underscored the need for understanding different communities in London's schools: "We were impressed by the risks being run in London if some of the lessons of New York were not learned... "Any form of colour blindness... would not work" (Newsam 2014, p. 72). Instead, all programmes and materials were designed to consider the student population that made up the schools of the ILEA. This included British Caribbean and Black British students, who in the 1970s were an increasingly large part of the overall student body.

Films produced by the ETS often included Black students as part of the opening credits, and some account of cultural differences was taken when presenting material. One example of this is the 1978 programme "Sing a Song" for primary school students. The opening credits of this programme feature a young Black girl playing on a seesaw in a classroom ("Farewell to the Network" 1979). The six programmes in series one were made with children of different racial backgrounds from Moorfields and St. Mary's R.C. Primary Schools. In addition to the television programme, teachers could also access learning materials connected with "Sing a Song", including cassette tapes and printed materials.

*Sing a Song Bumper Book One* (Bird et al. 1978) includes musical notation for the songs in the programme; however, it is not just a book of piano music. In both visual and written content, *Sing a Song* embraces the principle of multiculturalism. Like the opening credits of the television programme, the front cover of the book depicts children of multiple racial backgrounds playing together in harmony. Illustrations by Robin Mukerji show similar scenes of multiracial children playing and singing together, and these are complemented by photographs at the beginning and end of the book from the television programme with some of the children who were filmed for "Sing a Song." However, the multiculturalism depicted in *Sing a Song* was curricular as well as visual. Included in the written collection are three songs that connect, directly or indirectly, to Caribbean culture. One of these three, "Christmas Calypso," exemplifies the way that the ILEA made use of Caribbean musical forms.

Calypso was developed in Trinidad from traditional West African forms of music and was popularised globally in the 1940s and 1950s by calypsonians such as Lord Kitchener, Lord Invader, and the Mighty Sparrow. Importantly, calypso music comments ironically on current events, often in a wry way, through the use of upbeat melodies and often using percussion or syncopation to accent politically charged lyrics. However, the calypsos in *Sing a Song*, though they are upbeat melodies, do not otherwise celebrate Caribbean culture or history and certainly do not contain what Amanda Bidnall calls "Authentic" content: "Authentic calypso... was nothing if not politically incorrect and sometimes downright disreputable" (Bidnall 2017, p. 113). "Christmas Calypso" is based on a Virgin Islands calypso called "Guava Berry" by Bill La Motta (who is not credited in *Sing a Song*). While "Guava Berry" is, like "Christmas Calypso", a Christmas song, it is about going door-to-door, bringing greetings, and asking for the traditional Christmas drink of guava berry rum. One of *Sing a Song*'s editors and the director of the ILEA Music Centre, Wendy Bird, took the music and rewrote the lyrics. The mild patois in the original song is replaced with standard English, and the activities and attributes of the holiday are decidedly British: "paper chains and holly" (Bird et al. 1978, p. 33) rather than rounds of visiting and guava berry rum. The illustration for the song shows a white child chef with a spoon. The credit does not indicate where the "traditional music" is. In short, unless a child is previously familiar with calypso as a musical form, "Christmas Calypso" would not help them understand the Caribbean definition of calypso in any way.

The lack of cultural information is also noticeable in the television episode of *Sing a Song* that includes "Christmas Calypso." The white presenter, Tom Saffey, does not introduce calypso as a musical form or otherwise explain where the song originated, simply stating that the song is "all about what you have to do at Christmastime, making decorations, putting things up . . . mixing the Christmas puddings and baking your cakes, and generally making the house bright and sparkling" (ILEA TV 1976). He plays the song on the guitar while he sings with the children. The only percussion is a set of jingle bells, shaken by one of the children. These are not played on the downbeat, as they would be in Caribbean calypso, and the pace of the song is much slower than "Guava Berry", making it difficult to hear the resemblance.

By using traditional songs and musical forms such as calypso, *Sing a Song* goes beyond a mere pictorial representation of Black British children who might experience the ILEA's television programmes and associated material. However, the lack of commitment to cultural accuracy—the use of white British versions of Black Caribbean materials—suggests that the ILEA, whose "policy makers, officers, and teachers were mostly white" (Olowe and Minhas 1990, p. 11), either did not understand or were not concerned with including the heritage of their Black British students. As Black British sociologist John Solomos writes, multicultural policy in schools "was a means of incorporating the teaching of 'black studies' within the schools; however, at the same time depoliticising its content" (Solomos 1988, p. 81). Despite the good intentions of white liberal teachers and educators in the ILEA, multiculturalism in the ILEA's curriculum often results in only surface-level

acknowledgement of Black students and their heritage, rather than increased musical understanding for all its pupils.

### 3. Poets in School: Listening to the Voices of Young Black Students?

ILEA Television was not the only initiative trying to address its new population. The ILEA also had its own publishing arm, which produced, among other things, collections of student work. Although the ILEA published all forms of student writing, including essays, playscripts, and short stories, it most often published young people's poetry, something that was emphasised in this period because, as Harold and Connie Rosen put it at the time, poetry was a way of "using written language for their own purposes and of maintaining confidence in their own 'voices'" (Rosen and Rosen 1973, p. 92). In 1969, Alasdair Aston, a white Scottish poet and naturalist, founded Poets in School. Although the programme brought working poets such as Adrian Henri and David Harsent into school, Aston hoped that the students' work in response would "demonstrate that the poets in schools can be the pupils" (Aston 1977, p. xv). To that end, he produced a collection of young people's poetry written in response to the poets that was published by the ILEA, *Hey Mister Butterfly* (1978). This collection included some poetry by Black British students, and these represent Aston's and the ILEA's attempts to listen to the voices of young people. The choices of Black poetry made by Aston, however, indicate a preference for the same kind of multiculturalism found in the ILEA television programmes being produced at the same time.

Listening to student voices, according to Rachel Sophia Phillips, can have an especially powerful effect on at-risk students: "by simply increasing and incorporating student voice in schools, those who are alienated will often reengage due to a sense of ownership over what they are doing" (Phillips 2013). As Bernard Coard had established in the early 1970s, using the ILEA's own data from a report entitled *The Education of Immigrant Pupils in Special Schools for Educationally Subnormal Children* (1967), Black children were among the most at-risk students in London schools. Coard wrote in *How the West Indian is made educationally subnormal in the British school system* (1971) that too many Black children were being placed in educationally subnormal classrooms and that "Black children, who are young and unsure of themselves, may simply accept the judgement of themselves as being of low intelligence and give up any attempt to succeed academically" (Coard [1971] 2005, p. 32). The ILEA tried to respond to Coard's criticism through efforts such as Aston's Poets in School programme, hoping that listening to Black student voices and encouraging them intellectually would make a considerable difference in outcomes.

Aston wrote in his introduction to *Hey Mister Butterfly* that "I have tried to choose poems which are generally enjoyable . . . poems which explore or celebrate individual feelings or thoughts about particular circumstances" (Aston 1978, p. 5). However, as a naturalist, Aston tended to select poetry that was "generally enjoyable" to him—that is to say, nature poems. There are poems about all the seasons (including multiple poems, inevitably, about "Spring"), the sea, the weather, animals, and plants, and these as a whole dominate the collection. Aston's desire to "celebrate" and include "enjoyable" poems often means that the work chosen for the collection fails to reflect negative "feelings or thoughts about particular circumstances". For example, Robert Murray, aged 12, wrote "Caribbean Gem," which describes Jamaica as "a lovely gem" where people "laugh all day" (Aston 1978, p. 129). Leroy Tracy, aged 10, wrote a similarly unnuanced description of the country in "All About Jamaica", "a sunny land" (Aston 1978, p. 171) filled with good things to eat.

Even in poems about experiencing racism, Aston's choices are ambiguous. Donovan Parnell, aged 11, writing about "My Face," creates a series of statements about his face. He begins by stating that his face is "very unhappy" (Aston 1978, p. 139) and "always having trouble with the teacher" (Aston 1978, p. 139) in the first two lines; it is not until the fourth line (out of eight) that he reveals that his face is Black and not white. Because the lines always begin "My face is," it is not clear whether Parnell intends to connect his Blackness with his unhappiness or if these are separate features of his face. This ambiguity

is compounded by the poem's concluding lines, where his face is "funny" and "is as it is" (Aston 1978, p. 139). It is difficult to know if Aston had other poems to choose from that might have described the Black experience differently, but certainly most of the poems that are identified by young Black poets tend to avoid an entirely negative depiction.

One other noticeable feature about these poems is that they are all in "standard" English rather than patois, creole, or any other Caribbean form of English. This suggests either a concern by teachers to reward British versions of English or a desire on the part of the students to succeed in the education system, since students were often placed in ESN classrooms because "teachers cannot understand" the "kind of 'plantation English' which is socially unacceptable and inadequate for communication" (National Association of Schoolmasters 1969, p. 5), according to the National Association of Schoolmasters. Only one poem in *Hey Mister Butterfly* is in patois, and that is nineteen-year-old Lespaul Mackay's "Dreads in the Alley." The poem details Rastafarian "dreads" set upon by police and sent to prison for "SMOKING UP THE CALLY" (Aston 1978, p. 108); the Cally is a neighbourhood in Islington, North London, where Mackay went to school. Mackay never mentions the police directly, substituting the adjective "wicked" for the noun "police" as in "wicked sneak up quiet" or "WICKED SEND DEM TO PRISON" (Aston 1978, p. 108). Aston wrote in his introduction that he "tried to be sensitive to writers who have been inventing forms of expression suitable for specific experiences" (Aston 1978, p. 5), and—although Mackay had hardly invented either the "call and response" form of the poem or the patois used to express it—"Dreads in the Alley" certainly is a unique mode of expression in Aston's collection. It is also a rare example of political poetry in material produced by the ILEA's main publishing arm; it would be left to Black British educators working with the ILEA to thoroughly embrace young Black students' voice and politics.

## 4. Black Culture Supplement: Ansel Wong, the Ahfiwe School and the ILEA

The ILEA continued to produce material that emphasised positive multiculturalism, referred to by Troyna and Williams as the "saris, steel bands, and samosas" approach (Troyna and Williams 1986, p. 24). However, as Black educator Maureen Stone commented, "While schools try to compensate children by offering . . . steel bands, black parents and community groups are organizing Saturday schools—to supplement the second-rate education which the school system offers their children" (Stone 1981, p. 11). One of the community members heavily involved in this organisation was Ansel Wong, and his successful work would eventually be noticed by the ILEA.

In 1965, twenty-year-old Trinidadian Ansel Wong arrived in Britain to attend university. His training as a teacher coincided with rising concerns about British schools' inability to adequately integrate young Black students into classrooms. Wong was particularly interested in the work of Bernard Coard. According to Coard, the British schools, including the ILEA schools, led to the Black child acquiring "two fundamental attitudes or beliefs as a result of his experiencing the British school system: a low self-image, and consequently low self-expectations in life" (Coard [1971] 2005, p. 47). Coard's pamphlet, published by John La Rose's independent Black publishing venture, New Beacon Press, cemented in many Black British educators' minds the need for supplementary schools to help students escape ESN classrooms but also strengthen their identities through a deeper understanding of Black history and culture. Ansel Wong, influenced by Coard's writings, began his teaching career at the ILEA's Sydenham Girls Secondary School; by 1974, his experiences led him to ask Lambeth Council and the ILEA to fund and support a supplementary school in Brixton, the Ahfiwe School.

Like other supplementary schools, Wong's Ahfiwe School taught basic skills and Black history. It was not strictly an after-school programme or a Saturday school, like many supplementary schools; however, an alternative school that welcomed students who had not succeeded in mainstream classrooms. Ahfiwe was an alternative both to traditional schooling and to ESN classrooms; Wong told Zara Dalilah that the Ahfiwe School "was a full time school for young people who were truanting in the head or truanting with the feet

in the education system" (Dalilah 2017). Wong was an unusual choice to lead such a school for the ILEA, with his radical connections to the Black Liberation Front; however, he had worked with many community organisations to alleviate problems in the Brixton area with some degree of success. This, combined with his successful teaching at Sydenham and the ILEA's need to reach failing students in Brixton, led to a one-year trial appointment in 1974.

Wong, like Farrukh Dhondy, saw education as a cultural weapon. Black students were disaffected by teaching and a curriculum that ignored them. He made his mission the "blackening of the curriculum" (Waters 2019, p. 133) in all aspects—from changing mathematical story problems to include Caribbean or Black history examples to bringing performance poets to the school. However, Wong's main contribution to the creation of a Black curriculum that would strengthen Black youth identity was in his English curriculum. Rob Waters notes that "Black self-narration was central to the project of empowering pupils at Ahfiwe . . . [and] the process of narrating the black self [was] a dialogic one, in which black radical literature would play a formative role" (Waters 2019, p. 148). Students read a wide variety of politically-based writing, from Mao Tse-Tung to Vladimir Lenin, but focused on Black radical and activist writing by people such as Frantz Fanon, Eldridge Cleaver, and Angela Davis. Wong encouraged students to interact with these writers through forms of literary self-expression, poetry, essays, and autobiographies. These were published in the school's journal.

The emphasis on self-expression often brought Wong into conflict with the ILEA because the content of student writing demonstrated their knowledge and appreciation of radical Black thinking. This did not mean Wong was not interested in improving the educational achievement of his students. As he said later,

I wanted to have an educational initiative that stood in its own right, that wasn't a supplement. It had to be mainstream. Aspects of that had to be its culture, history, heritage, language and all sorts. So there would be homework sessions, there would be opportunities in terms of basic skills, and there would be mainstream educational provision on African history, Caribbean history, dance, drama, and all that sort of thing (Waters 2016, p. 27).

The ILEA at first encouraged this type of teaching as a way of encouraging disaffected pupils. However, as with the ESN schools, Wong's Ahfiwe school did not return many of its students to mainstream education, and the ILEA withdrew funding because, as Wong suggested to Rob Waters, "the very words 'black studies' were anathema to them. We felt that that was important, but they said no, that is not the purpose of the school, the school is basic education" (Waters 2016, p. 28). However, student writing as published in the *Ahfiwe Journal* suggests that students not only achieved basic education but went beyond it to see the value of education as a means to a career. Evelyn Christie, a student at Ahfiwe, wrote an essay for the school publication entitled "Lissen Sisters," encouraging Black girls to stay in school. In it, she commented, "I hear a lot of girls saying to hell with 'A' level, 'O' level, CSE, and GCE, but it will help get you somewhere in life [. . .] We should have ambition, education, and a career in front of us" (Christie 1974, p. 9). Laura Tisdall suggests that Black girls had to "plan carefully if they did not want their futures to be derailed by the racist stereotyping they experienced in school" (Tisdall 2022, p. 499). By leaving mainstream ILEA schooling and going to the Ahfiwe school, Evelyn Christie was able to escape the racist stereotyping; however, ILEA officials did not recognise that she was also gaining skills in writing, reading, and critical thinking.

Evelyn Christie's essay showed the value of education for her. However, the ILEA's actions showed that they were less concerned—whatever they said—with educational improvement than with the potential radicalisation of the students. This is shown most clearly in the success of student poets like Janet Morris. Morris, a student at Wong's school, published her work in the *Ahfiwe* journal; however, she simultaneously became a published poet outside of the school, bringing national attention not only to her but to the Ahfiwe School. Like many young poets at the school, Morris wrote about the experience of being Black in Britain. Wong published her poem "Black" in the first (1974) edition of the *Ahfiwe* journal; the poem is an indictment of the white-run system of enslavement that continues

to affect Black people in the present day, suggesting that white people's words result in the death of "our brothers" (Morris 1974, p. 5). But importantly to the stated goals of the ILEA, it is also a well-crafted poem: a sestet in two three-line stanzas, with the first lines of each stanza paralleling each other (one about "the black man" and one about "the white man") and the last two lines of each stanza containing end rhymes. This poem, along with two others, were published in the girls' magazine *Petticoat* and received a sum of £24 after they "were passed on by a friend at the educational project" (Smith 1975, p. 1), possibly Wong himself. However, it was another of Morris's poems, passed on by the same friend, that caused her to receive attention in the national press. This poem, "Babylon," was submitted to the monthly journal of the Lambeth Community Relations Committee, which published it in the late spring of 1975.

"Babylon" is about the constant pressure Black people in Britain face from the police. It is certainly not complimentary to the police, but it is also an expression of Black pride. Morris writes, "The day will come when we'll be strong/To fight the babylon back" ("Baez to Babylon" 1975, p. 6). The poem caused an uproar when Sir Robert Mark, Metropolitan Police Commissioner, complained that it was a 'thinly veiled incitement to confrontation' (Smith 1975, p. 1). Two weeks later, Sir Frederick Bennett, Conservative MP for Torbay, was calling for Morris's prosecution under the Race Relations Act of 1965 for incitement to racial hatred (Cole 1975, p. 18). While the attorney general declined to take the matter further, the incident caused the ILEA to rethink its funding for Ahfiwe, and the school faced closure. Just as with the watered-down "Christmas Calypso," the radical content of the poem was more important than its form; however, the ILEA did not have the power to change the words of Black students as it had with Black Calypsonians.

The ILEA attempted to support other supplementary schools, but these also closed. Black educationalist Maureen Stone suggests that while supplementary schools for Black children often had "help and encouragement from a local primary Head" (Stone 1981, p. 184), these very supporters could remove their support "when black children who were attending started criticizing teachers for not telling them about Black History" (Stone 1981, p. 184). The ILEA's leaders felt that by removing Black students from traditional classrooms, they were allowing for radicalisation that encouraged Black youth to try and overturn the system. The ILEA closed Ahfiwe School in 1976. However, Wong's experiment was not without effect, as it was part of the ILEA's decision to move away from nonconfrontational multiculturalism and toward anti-racist policies in mainstream classrooms in the inner London boroughs.

## 5. Achieving Anti-Racism, Losing the ILEA: Len Garrison's ACER

Anti-racism differed from multiculturalism in that, beyond simply supporting and embracing the cultures and heritages of students, it also required individuals and institutions to examine structural inequalities that acted as barriers to achievement for Black and other ethnic minority pupils. In the mid-1970s, racial tensions in London were increasing, and anti-racist protests were increasing as well among both Black and white Londoners. This was especially true in schools; organisations such as All-London Teachers Against Racism and Fascism (ALTARF), the National Association of Multicultural Education (NAME), and School Kids Against Nazis (SKAN) were among many groups made up of both Black and white pupils and teachers operating within London schools (Sands-O'Connor 2018).

However, many teachers, while supporting anti-racist initiatives in principle, did not know how to enact them in their own classrooms. Tuku Mukherjee suggests that British schools had dealt with racism through multiculturalism, which had "been constructed in such a way that the issue is perceived as having nothing to do with white teachers, white institutions and white power structure" (Mukherjee 1984, p. 182). Anti-racism, on the other hand, demanded a commitment to the idea "that the roots of racism lay in the nature of British society and its imperialist past and that unless the social and economic base of society were transformed, there would always be the possibility of racism" (Doherty 1984, p. 217). True anti-racism would require dismantling the education system. The ILEA's

attempts at doing this with Wong's Ahfiwe school proved unsuccessful, partly because it only dismantled the system for Black students. The ILEA wanted something that could reach all students, as their television programmes had; however, with an anti-racist stance. They labelled this approach "education for a multi-ethnic society." In a pamphlet explaining this approach, the ILEA told teachers,

> Schools play a unique role in helping to shape the attitudes and understanding of the rising generation. In our multiethnic society, a principal aim for schools must be to develop in children values which are consistent with and appropriate for that society—that is, an interest in, and respect for, diversity and a fundamental commitment to equality for all. (Cocking 1984, p. 10)

Racism, the ILEA pamphlet argues, "undermines and negates these objectives" and therefore is "an important educational issue in all schools, whether they are ethnically mixed or not" (Cocking 1984, p. 10). Len Garrison, a Black educator and later founder of the Black Cultural Archives, was one of the people that the ILEA approached to help them.

Garrison had already started the Afro-Caribbean Educational Resource (ACER) project in 1976 to create "a major Afro-Caribbean library resource base as well as develop a programme of materials which identified the ethnic, cultural and social dimensions of the black child" (Garrison 1990, p. 175). The ILEA liked Garrison's approach, which involved "a resource list for teachers wanting to make contact" with "black and anti-racist educators, writers, publishers and campaigners" (Waters 2019, p. 151) as well as creating new educational materials for teachers and students and providing a lending library of works by Black writers and educators. Garrison's stated philosophy also aligned with the ILEA's ideas about education for all students:

> [ACER's] philosophy was that schools should be made to work for all pupils. Black children should not have to go outside the mainstream schools to seek recognition and gain confidence in being themselves. White pupils should also learn about being black in a white society. (Garrison 1990, p. 175)

ACER and the ILEA formed a partnership in 1977, and it lasted until 1988, one of the longest partnerships that the ILEA had with a Black educational group.

Wong's Ahfiwe school ultimately had not succeeded because the ILEA claimed it failed to address basic skills while pushing a Black Power agenda—both of which, according to the ILEA, further alienated Black students from the educational process. Garrison succeeded by providing learning materials for mainstream classrooms, in much the same way that the ILEA's own television service had, and promoting the benefits of cross-cultural understanding. In a time when tensions between Black communities and institutional Britain were on the rise, ACER's offer to the ILEA seemed not only a good fit but an increasingly necessary intervention.

However, like Wong, Garrison was more interested in empowering Black students than white students. In 1977, just before Garrison formalised the partnership with the ILEA, he travelled to Nigeria for the Commonwealth Festival of Arts and Culture (FESTAC77) and presented a talk on "Black Youth, Rastafarianism and the Identity Crisis in Britain." This talk, which was later expanded and published by ACER, argued that "black youth in Britain today find themselves torn between two cultures and two sets of values and expectations and alienated from both" (Garrison 1979a, p. 13). Partnering with the ILEA allowed Garrison to address Black student identity while at the same time giving them a reason to remain in mainstream education.

Like Wong, Garrison encouraged young Black people's self-narration. He ran creative writing classes, using prompts that demonstrated his understanding of young Black students' interest in Black Power, Rastafarianism, and anti-police protests. He also encouraged them to write in a language that was comfortable to them, and many chose to write in patois. This was crucial because, as Bernard Coard pointed out in his treatise on why many Black children were placed in ESN schools, "The West Indian child is told on first entering the school that his language is second rate" (Coard [1971] 2005, p. 45). The child who wanted

to avoid ESN classrooms had to, as Baroness Floella Benjamin did when she attended London schools in the 1960s, "learn to speak the Queen's English" (Benjamin 1997, p. 100). However, Ansel Wong had already recognised through his work with the Ahfiwe school that for many Black students, "Patois is the principal and private bearer of cultural capital. It is one way of asserting individuality, an assertion that, in its aggressive expression, often invites responses of annoyance and dislike" (Wong 1986, p. 120). Dick Hebdige calls patois "that shadow-form, 'stolen' from the Master" (Hebdige 1979, p. 31), powerful because it was partly but not wholly recognisable to white British people. Garrison, by allowing (even encouraging) students to write in patois, was offering a new definition of what was educationally normal.

Garrison wanted teachers to see the writing of Black students as valuable. In his teacher guides, such as *Images and Reflections* (Garrison 1979b), he includes previously-published poems by Black youth and children, such as "I am a Little Black Girl" (Garrison 1979b, p. 3), originally published in ten-year-old Accabre Huntley's collection, *At School Today* (1977), and Hugh Boatswain's 'I Remember' (Garrison 1979b, p. 23), originally published in community-based publisher Centerprise's *Talking Blues* (1976). Perhaps deliberately, these and other poems that Garrison chose for the teacher's guide are in standard British English, although both Huntley and Boatswain sometimes wrote using patois words and phrases in other poems. These two poems connect identity to heritage; Boatswain looks back to the Caribbean where he was born, "Buying two cents worth of sugar cake" (Garrison 1979b, p. 23) before school; and Huntley's poem looks back to an African heritage. However, both poems also express a desire for African and Caribbean history and heritage over a British one: Huntley says she will sing an African song "all day long" (Garrison 1979b, p. 3) despite being in England, and Boatswain, after detailing a school experience where he learned British history but "nothing/Of life in our own society" adds, "I was not angry then./I did not understand" (Garrison 1979b, p. 23). These poems aim to demonstrate to teachers in the ILEA the importance of providing history and experiences that belong to all students in London schools. Garrison's selection of poems written in standard English provides teachers with the assurance that students who learn their own history in school will not reject other, more mainstream teaching.

Garrison put his philosophy into action by creating a Young Writers Competition for Black youth, many of whom he tutored in his own creative writing courses. His definition of "youth" was broad and representative of the population that the ILEA served; when a book of the winners of the competition, *Black Voices*, was published in 1987, it contained poems and essays from writers as young as nine and as old as thirty. The writing prompts were sometimes used as poem or essay titles, as repetition of titles such as "What makes a Black person Black?" or "Let the pen speak for I" throughout the book indicates. Garrison's embrace of youth interest in Rastafarianism is also clear in these prompts, as there are several that allude to Rastafarian ideas. Winning essays from Rastafarian youth stress an emphasis on peace; sixteen-year-old Janette Williams writes, "Rastas believe in peace among everyone" (Williams 1987, p. 79), and nine-year-old Dean Stewart announces he is a Rasta before adding, "We must all live together in peace and feel secure praising Jah more and more" (Stewart 1987, p. 71). While there is an occasional complaint about police violence, it tends to be from older writers; twenty-four-year-old Benson Idowu complains that "agents of the law forces [sic] their muscles by arbitrary arrest of Black youth under the obnoxious SUS laws" (Idowu 1987, p. 197). But importantly, these sentiments are expanded upon in the teacher materials created by Garrison. In his introduction to *Images and Reflections*, Garrison writes, "Why do black youth constantly experience harassment in their urban communities? Why is the education system failing to respond to the changing needs of its pupils?" (Garrison 1979b, p. 2). The material goes on to include information on patois and sus laws, with the admonition that "The extent to which white attitudes have been formed by an education full of myths and misrepresentations—inheritance of colonialism and imperialism—must persuade teachers, however overworked and under pressure, to recognise the necessity for change" (Garrison 1979b, p. 2). Garrison tried to

show teachers that Black anger was the result of white indifference and miseducation; he gave them the material to educate themselves; however, teachers had to decide whether it mattered.

Proof of Garrison's commitment to young Black writers can be found in the testimonials of his students, who praised the way he worked with them to challenge cultural stereotypes of Black people and to become involved in political change (Sands-O'Connor 2022, p. 137). However, when one of his young proteges, Elaine Clair, died under mysterious circumstances, Garrison was deeply affected. Clair's imaginative response to the prompt, "How do you see yourself in the year 2001?" won the overall prize in the year of her death, 1981. In it, she wrote,

> I have a hope and ambition for my people and all peoples. I want life to be good again. I want society to realise that all peoples, irrespective of colour, religious background or political ideologies can live together: the aim is to blot out racism, and work together. (Clair 1987, p. 330)

Clair had learned the lessons of anti-racism and self-narration, but she disappeared and was found drowned in the Grand Union Canal two miles from her home. She had sustained bruises to her head and neck, but the police declined to investigate. Garrison wrote an elegy to her in which he described her as "a young warrior" (Garrison 1985, p. 56). Although the ACER Center and the Writing Competition continued to operate until the ILEA was dissolved in 1988, Garrison took a step back from ACER and began focusing his attention on what would become the Black Cultural Archives. Garrison's attempts to work within the system without giving up his politics ultimately failed. The Writing Competition became less centred on Black *British* youth, issuing prompts concerning Martin Luther King's "I have a dream" speech and apartheid in South Africa; the second collection of winning essays contained no Rastafarian prompts. ACER without Garrison tried to take a step back from Black politics, but ultimately, this turn away from challenging British institutional structures would not save the ILEA from governmental accusations that it was allowing the radicalisation of Black students through its literature and writing offerings.

## 6. Conclusions

In 1980, the former head of Birmingham children's library services, Judith Elkin, could write that Garrison's ACER project "analyses the identity crisis facing black youth today and looks at ways in which education needs to take a positive stand to affirm cultural differences" (Elkin 1980, p. 18). Educator Gillian Klein recommended ACER's poetry collections in 1985 as "examples of emotive poetry in dialect by young people" (Klein 1985, p. 102). By 1986, Baroness Cox was arguing in the House of Lords that the ILEA's anti-racist efforts, including some of the material in the lending library at ACER, "are highly politicised and appear designed to destabilise our society" (Hansard 1986). Although teachers and librarians in and out of the ILEA praised the attempts to support London's growing Black student population and promote anti-racist attitudes, the government saw the ILEA as dangerous and initiated legal procedures to abolish the ILEA. Britain moved toward a national curriculum, one that Black parent Conrad McNeil called "not truly national. The culture reflected in it is that of the prevailing Anglo-Saxon persuasion which excludes the significant input from the Caribbean and elsewhere" (McNeil 1988, p. 10). Rob Waters argues in *Thinking Black* that "The black education movement was always inevitably bound up with the designs of the state" (Waters 2019, p. 163) and that relations between Black organisations and the state became strained over the issue of Black studies. The ILEA tried multiple approaches to including, celebrating, and strengthing Black voices; however, in so doing, it also gave space to voices that were critical of institutional efforts to contain the Black population. This created an existential crisis in the ILEA, which relied on government support, which resulted in its demise. Even as it tried to appease a right-wing government by softening (if not silencing) its Black partner voices, the ILEA was dismantled over fear of radical politics directing the curriculum. Black studies as a cultural weapon was dismantled along with it; however, the debate around the place of Black

people in Britain's now-national curriculum continues. From Michael Gove's attempts to remove Mary Seacole from the curriculum in 2013 (Sands-O'Connor 2016, p. 149) to the Centre for Literacy in Primary Education's annual *Reflecting Realities* reports (which first appeared in 2018) to the founding of independent charity The Black Curriculum in 2019 to address the lack of Black British history in schools, educationalists, government ministers, and Black activists continue to argue over how—and sometimes whether—Black people should be addressed in Britain's schools. The ILEA's creation of literature by and for Black students provides different models for how to achieve a curriculum that addresses all students; however, its demise suggests that organisations wishing to do so should consider the larger context.

**Funding:** This research received no external funding.

**Institutional Review Board Statement:** Not applicable, as this study did not involve human or animal subjects.

**Informed Consent Statement:** Not applicable.

**Data Availability Statement:** Not applicable.

**Acknowledgments:** Thanks are given to the British Academy and to the London Metropolitan Archives, who were willing to transfer the film of the ILEA's *Sing a Song* programme into a format that researchers like myself could access freely.

**Conflicts of Interest:** The author declares no conflict of interest. The funders had no role in the design of this study, in the collection, analysis, or interpretation of data, in the writing of the manuscript, or in the decision to publish the results.

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
