# Peer review of "“Education Is a Cultural Weapon”: The Inner London Education Authority and the Politics of Literature for Young People"

_humanities, doi:10.3390/h12050109_

Round 1

Reviewer 1 Report

The article could benefit from a more explicit exploration of the methodologies employed to gather and analyze historical sources. This would provide transparency and enhance the credibility of the research process. Additionally, the article might consider incorporating visual aids, such as charts or timelines, to help readers visualize the chronology of key events and shifts in educational policy.

The conclusion offers a succinct summary of the central arguments and effectively ties together the themes explored throughout the article. The notion that the dismantling of Black studies paralleled the demise of the ILEA as a whole is a thought-provoking insight, inviting readers to reflect on the lasting impact of institutional decisions on educational equity and diversity.

In summary, "“Education is a Cultural Weapon”: The Inner London Educa
tion Authority and the Politics of Literature for Young People" provides a compelling examination of the challenges faced by the ILEA in its pursuit of promoting Black studies. The article successfully weaves together historical perspectives, educational policies, and political dynamics to offer a nuanced understanding of this complex issue. With some minor enhancements in terms of context-building, methodological transparency, and visual aids, this article has the potential to make a valuable contribution to the field of educational history and policy analysis.

Author Response

Thanks for your comments!  I added some contextual material at the beginning so that readers can see how the essay was developed, and also be signposted to some of the "facts and figures" that are much better examined in educational reports.  Shifts in educational policy were not entirely clear-cut, even within the ILEA, so it would be difficult to include a timeline.

Reviewer 2 Report

"Education is a Cultural Weapon" is a fine and engaging article.

The introductory quote -“Education is a cultural weapon and to black people, culture means political freedom”-is provocative and invites the reader to want to know more.

The narrative presents the stage for the body of the text along with a fine use of direct quotations form the students' work in the ILEA schools. The honesty presented in the narratives expressed in the student's poetry clearly challenged the school offices, and rightly.

It would be interesting to know what has come of some of these students/now adults. 

Author Response

Just for reference, I did address what happened to some of the young Black writers in my book, British Activist Authors Addressing Children of Colour, but didn't want to take the focus away from the ILEA here.  Thanks for your comments!

Reviewer 3 Report

I loved the article. Really useful and engaging. I worked in London in the mid-1980s and this resonated at a personal level, as well as academically. Thank you.

Author Response

Thanks so much for your comments!

Reviewer 4 Report

This is a fascinating, well written and well researched account of the history of the ILEA and its developing approaches to education in inner-city London. The paper clearly sets out the struggles of the ILEA in serving its diverse population and the criticisms of its, at times, tentative approach. By doing so, it might be underplaying quite how radical some of the approaches were at the time. I wonder whether a bit of context in terms of both some data on the proportion of Black teachers, Black pupils, levels of attainment, etc would be beneficial, in addition to a discussion of how the terms multi-culturalism and anti-racist approaches were seen at the time. 

In the conclusion, it may be worth providing some link to where we are today in terms of an anti-racist education and a curriculum that reflects a diverse population and provides minority ethnic children with an understanding of their history within Britain.

This is a minor point but I wonder whether the discussion of a 'lack of violence' in the writing of Rastafarian youth is better described as an emphasis on peace? On the same page, there is capital I missing in line 459.

These comments are really for the editors but just to make you aware that they were picked up:

There are some issues with the typesetting on p1 and 2 and the typesetting of the references.

Author Response

Thanks for your comments!  I have provided some context at the beginning and also signposted to where we are now in the conclusion.  I also liked your change regarding the writing of young Rastafarians to emphasize peacefulness, so I made that change as well.